# The role of scenic context on upright face preference in infancy

**Megumi Kobayashi** [1]*, **So Kanazawa**[2], **Masami K. Yamaguchi**[3]

**1** Department of Psychology, Niigata University, Niigata, Japan, **2** Department of Psychology, Japan Women's University, Bunkyo City, Japan, **3** Department of Psychology, Chuo University, Hachioji, Japan

* mkobayashi@human.niigata-u.ac.jp

**Data Availability Statement:** All relevant data are within the paper and its Supporting Information files.

**Funding:** This research was supported by a Grant-in-Aid for Scientific Research on Innovative Areas,

## Abstract

Scenic information plays an important role in face processing, whereas it has received limited attention in the field of developmental research. In the current study, we investigated whether infants, like adults, utilize scenic information for face processing by the preferential-looking method. In Experiment 1, we examined 4–5 and 6-7-month-olds' visual preferences for upright faces compared to inverted faces in two surrounding scene conditions: intact (in which a face occurs in an intact scene) and scrambled (in which a face occurs in a jumbled scene). We found that 6- to 7-month-olds preferred the upright face in the intact scene, but not in the scrambled scene. Meanwhile, 4- to 5-month-olds showed significant upright face preference in both scenes. The results of Experiment 2 ruled out the possibility that the lack of preference for upright faces in the scrambled scene in 6- to 7-month-olds resulted from more distraction by the scrambledness of the image than occurs with 4- to 5-month-olds, by showing no developmental changes in preference either for the scrambled images or the intact images when faces did not appear. Our results suggest that infants aged 6 months or more utilize scenic information for face processing.

## Introduction

Face processing becomes more sophisticated with increased age. Newborns, to some extent, have the ability to process faces showing a preference for upright facial photographs and face-like schematic configuration than their inverted counterparts [1–7]. Infants aged older than 3 months, however, show a preference for upright faces only for realistic, natural, or complex face stimuli, not for face-like schematic configurations [8–14]. These recent findings imply that preference for upright faces depends on face-specific mechanisms and/or more developed facial representations in infants older than 3 months.

The infant studies described above have presented faces only against a uniform background in isolation from a visual scene or context to investigate the upright face preference. Infants in the real world, however, see and learn faces in a complex, natural setting. Considering that perceptual experience provided by natural environments affects the development of face processing (e.g., "perceptual narrowing" phenomenon; for reviews see [15, 16]), we raise the possibility that scenic context plays an important role in face processing in infants.

'Construction of the Face-Body Studies in Transcultural Conditions' from the Ministry of Education, Culture, Sports, Sciences and Technology (MEXT) KAKENHI (17H06343) and a Grant-in-Aid for Scientific Research from JSPS for MKY (26285167) and MK (16H07424, 23H01055) for MK. The funders had no role in study design, data collection and analysis, decision to publish, or preparation of the manuscript.

**Competing interests:** The authors have declared that no competing interests exist.

Little work examined infants' face processing in cluttered visual displays and showed that infants preferentially looked at a face embedded among multiple non-face distractors [17–19], or from realistic scenes, e.g., animated films or natural scenes [20–23]. For instance, Amso and colleagues [22] found that attention to faces in the first second of scene viewing was shown both in 4-month-olds and adults during free viewing of natural scenes that include people. Also, even 3-month-old infants detect and fixate on a face of a person in a naturalistic scene [23]. However, no study examines the effects of natural scenic contexts on upright face preference in infancy. In the current study, we examined the effect of context in natural scenes on the upright face preference in infants aged between 4 and 7 months who are developing more sophisticated face processing.

In adults, some studies examined face processing in natural scenes. Previous studies suggest that face detection relies not only on a simple skin-colored, face-shape template that has a natural height-to-width ratio [24–26] but also on natural scenic contexts [27–29]. For instance, to test the effects of scene information on face detection, Lewis & Edmonds [27] asked participants to detect a face that appeared in an intact natural scene and a scrambled natural scene. Results showed that participants detected a face in the intact natural scene faster than in the scrambled scene. According to a face detection model proposed by Lewis & Ellis [29], scenic context facilitates the pre-attentive or early attentive stages of face detection by directing the observer's attention to candidate locations containing the face. Their findings demonstrate that face detection is not just the finding of a face, but also includes an analysis of the entire visual scene for clues about the location of the faces [28, 29].

Lewis & Ellis [29] imply that the context in which a face always appears on top of a human body plays an important role in face processing. Consistent with this view, previous studies have shown that bodies are a clue for human detection in a natural scene, e.g., [30–32]. Under the free-viewing or gender discrimination task, participants' eye movements were recorded when observing two scenes (one contained a single person and the other did not) side by side [31]. Participants showed a strong bias towards the person-present scene than the person-absent scene. More importantly, the first fixation is directed more at the person's body than the face, and the second fixation is directed at the face. Then, participants fixated stably on the face. These results are replicated by Fletcher-Watson and colleagues [32]. Bindemann and colleagues [30] investigated the role of the face and body in person detection in a natural scene by an eye-tracking method. Participants fixated more quickly and slightly more accurately on the upper body than the face immediately after the scene appeared, then they fixated on the face preferentially, suggesting that while attention is captured by faces, this effect would be mediated by body information in a natural scene.

Our goal was to investigate the effect of context in natural scenes on face processing in infants aged from 4 to 7 months. To this end, we used a preferential-looking paradigm to examine infants' preference for an upright face versus an inverted face. The current study compared infants' upright face preference between two conditions: an intact scene and a scrambled scene. As used in the previous studies, e.g. [27], a face appeared in a normal intact scene in the intact scene condition, whereas a face appeared in a jumbled scene in the scrambled scene condition. We predicted that if the scenic information played a role in infants' visual preference for an upright face, the upright face preference should differ between the intact and scrambled scene conditions. On the other hand, if infants disregarded the scenic context when looking at a face, there should be no differences in infants' preference for upright faces between the two conditions.

We divided infant participants into 4- to 5-month-olds and 6- to 7-month-olds because we expected that scenic contexts affect face processing in 6- to 7-month-olds, but not in 4- to 5-month-olds for the following reason. As suggested by adult studies showing that the context

that a face appears above a body affects face processing [29–32], the development of body-specific processing and/or knowledge about the human body would be related to the current study. Previous studies showed that infants aged over 6 months acquired knowledge about human body structure and holistic body processing [33, 34]. Considering these previous findings, we predicted an interaction in the upright image preference between age and scene condition. That is, 4- and 5-month-olds would prefer the upright images in both the intact and scrambled scene, whereas 6- and 7-month-olds prefer them only in the intact scene.

## Experiment 1

In this experiment, we investigated whether scenic information affected the preference for upright faces in infants aged 4 to 5 months and 6 to 7 months. To this end, we examined infants' preferences for upright images compared to inverted images in both intact scene and scrambled scene conditions using a preferential-looking procedure. We expected that if infants utilized the scenic information in face processing, the upright face preference in the scrambled scene condition would be lower than that in the intact scene condition.

## Materials and methods

### Participants

The final sample of this experiment was thirty-five 4- to 5-month-old infants (14 females, *M* age = 137.60 days; range = 108–163 days) and thirty-seven 6- to 7-month-old infants (19 females, *M* age = 198.35 days; range = 167–223 days). As in the previous infant studies [35–38], infants aged from 105 days to 164 days were assigned to the 4- and 5-month-old group and infants aged from 165 days to 224 days were assigned to the 6- and 7-month-old group. They were all healthy and full-term Japanese infants. Within each age group, all infants were tested in both intact scene condition and scrambled scene condition.

An additional 48 infants were tested but were excluded from the analyses due to premature birth (N = 1), crying and/or fussiness (N = 23; 4–5 months: N = 11, 6–7 months: N = 12), a side bias greater than 90% (i.e., looking more than 90% of the time to one side) in either of the two conditions (N = 17; 4–5 months: N = 10, 6–7 months: N = 7), an experimental error (N = 1), or short looking time in either intact or scrambled scene condition (less than 36 seconds of total looking time, and the minimum threshold for each trial was 3 seconds out of 10 seconds) (N = 6; 4–5 months: N = 2, 6–7 months: N = 4). Participants were recruited through advertisements posted in local information magazines. Experiment 1 was conducted between February 2016 and February 2017. The study was explained to the parents and their written informed consent was obtained. Authors had access to information that could identify individual participants during or after the data collection, and the information was anonymized so that videos that recorded infants' looking behavior and looking time data contained no identifying information. This study was approved by a local ethical committee of Chuo University (2015–13, 2016–20).

### Stimuli

All stimuli used in this experiment consisted of full-color still images downloaded from the internet. Seven different natural scenes were selected, and each contained a human body and the face of a different Japanese female. The images were cropped to 309 pixels in width by 438 pixels in height using Adobe Photoshop. The faces within each image were about 146 (from hairline to chin) x 103 (face width) pixels. Each image was split up into nine cells (3 x 3 grid) by covering it with white-bordered rectangles. The face appeared in one of the three cells of

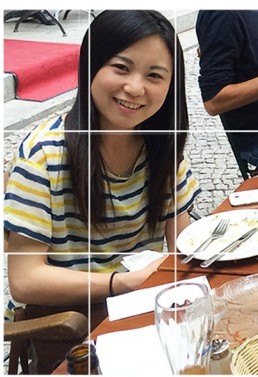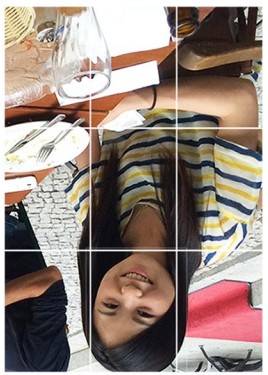
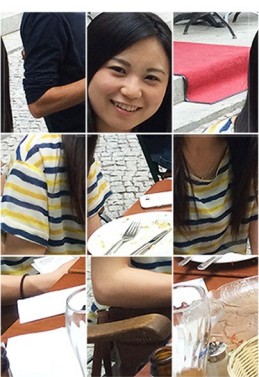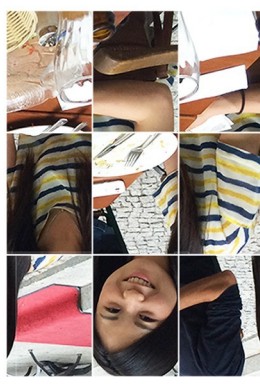

**Fig 1. Examples of stimuli used in Experiment 1.** Upper: the upright (left) and inverted (right) images in the intact scene condition where a face appears in an intact scene. Lower: the upright (left) and inverted (right) images in the scrambled scene condition where a face appears in a jumbled scene.

the top row, with a frontal view and a happy facial expression. For scrambled images, we rearranged the order of the three cells of each of the three rows, respectively. Thereby, the order in one row was different from those of the other two rows, and there were no two vertically adjacent cells from the original intact image. Inverted images were produced by rotating the whole upright image 180 degrees (Fig 1).

Each image subtended approximately 16.78˚ x 23.58˚, and the face cell in each image was approximately 5.44˚ x 7.72˚ when infants viewed the image from approximately 40 cm. When presenting upright and inverted images side by side on the screen, the distance between the inner portion of the faces was approximately 13.83˚. The background was a homogeneous white field.

## Apparatus

Each infant was tested while seated on his or her parent's lap and positioned at approximately 40 cm from a 21-inch CRT monitor with two loudspeakers, which was controlled by a computer (Dospara Prime Galleria). Both the infant and the monitor were located inside an enclosure made of iron poles and covered with black cloth. A charge-couple device (CCD) camera positioned just below the center of the monitor, which fed into a TV monitor and a digital video recorder, was located outside the enclosure. Throughout the experiment, each infant's looking behavior was recorded with the CCD camera. The TV monitor, which was connected to the CCD camera, displayed a live image of the infant's face to allow the experimenter, who was blind to the condition, to know when to start the sequence of the trial.

The parent of each infant was blind to the hypothesis of the study and the predicted direction of infants' looking behavior in each experimental condition. The parent was instructed not to look at the monitor and to remain silent.

## Procedure

Infants in each age group were tested using a preferential-looking procedure composed of six 10-s trials in each image condition: the intact scene and the scrambled scene. Three images out of 7 images were randomly selected for each participant in the intact scene condition, and the scrambled versions of the same three images were presented in the scrambled scene condition.

Every infant underwent six trials for each condition. The order of presentation of three images and the position of the targets (left or right) for each image in each condition were counterbalanced across participants across trials. Each trial began with a cartoon image appearing at the center of the monitor screen, with an accompanying beeping sound to attract the infant's attention to the monitor. As the infant fixated on the fixation cartoon, the experimenter turned off the cartoon image and presented the upright image and its inverted counterpart side by side. For each trial, the duration of stimulus presentation was fixed at 10 s regardless of whether infants looked: the total duration of each condition was 60 seconds. We defined the images in which the upright face appeared as the target. The order of presentation of the two conditions was randomized across infants.

Video recordings of infants' eye movements, which were anonymized, were analyzed frame-by-frame by an observer who recorded the total looking time for the left side and right side on the monitor in each trial. A second independent observer analyzed 25% of the total data for both the intact and scrambled scene conditions. No identifying information on looking-time data was included. Both observers were unaware of the image condition and the position of the target (left or right) on the screen. Interobserver agreement (Pearson correlation), as computed on total fixation times for the upright and inverted images across six trials, was $r = 0.93$, $p < 0.001$.

## Results and discussion

Table 1 represents the average total fixation time for the upright and the inverted images in the intact and scrambled scene conditions, respectively. To investigate the effect of age (4–5, 6–7), scenes (intact, scrambled), and orientation (upright, inverted) on the total fixation times for the upright and inverted images, we conducted three-way ANOVAs of the scene and orientation as within-participant factors and the age as a between-participant factor. The ANOVA showed a significant main effect of orientation ($F(1, 70) = 67.71$, $p < 0.001$, $\eta^2 = 0.27$. The significant two-way interaction between age and orientation ($F(1, 70) = 4.85$, $p = 0.031$, $\eta^2 = 0.02$) was significant, revealing that the total looking time for upright images was longer in 4- and 5-month-olds than in 6- and 7-month-olds, and that infants in both age groups looked longer to the upright images than to inverted images. We also found significant two-way interaction

**Table 1. Mean and standard deviation of total fixation time in 4- and 5-month-olds and 6- and 7-month-olds for the upright and inverted images in the intact scene and the scrambled scene.**

| | | Intact | | | | Scrambled | | | |
| | | Upright | | Inverted | | Upright | | Inverted | |
| | N | M (s) | SD (s) | M (s) | SD (s) | M (s) | SD (s) | M (s) | SD (s) |
|---|---|---|---|---|---|---|---|---|---|
| 4–5 m | 35 | 29.43 | 5.68 | 21.14 | 4.88 | 29.16 | 6.21 | 20.99 | 5.50 |
| 6–7 m | 37 | 28.38 | 4.84 | 21.29 | 3.29 | 25.65 | 5.64 | 23.24 | 5.05 |

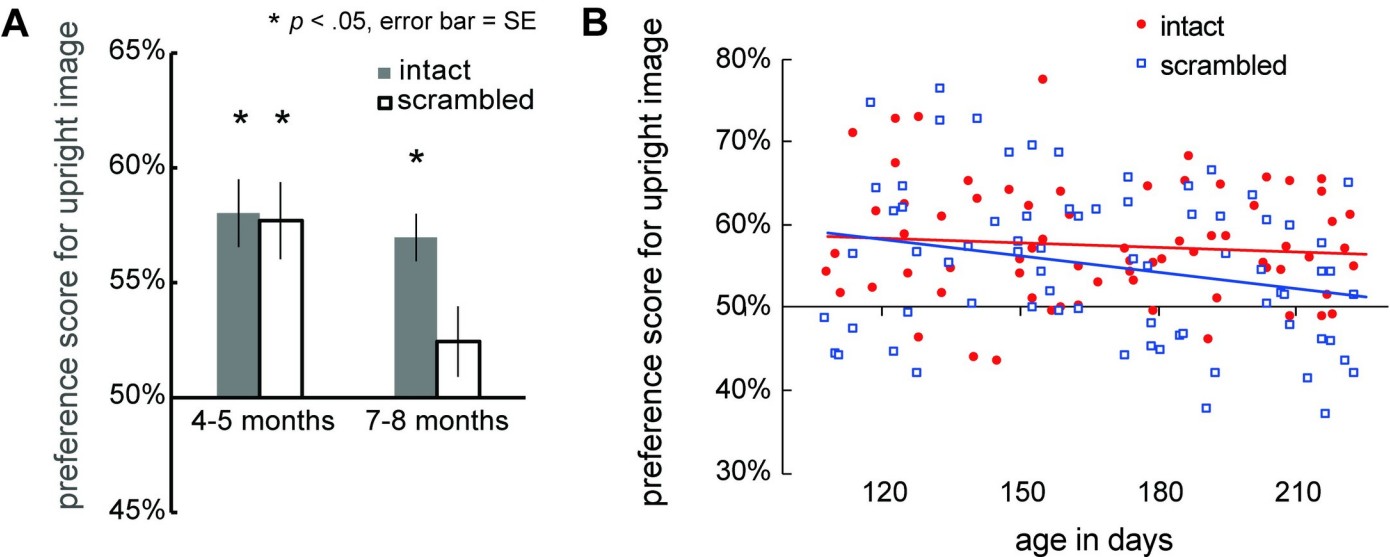

**Fig 2.** (A) Mean preference scores for the upright face in infants aged 4 to 5 months and 6 to 7 months. The gray bars indicate the upright preference in an intact scene, and the white bars represent the upright preference in a scrambled scene. The error bars show mean standard errors (SE). Significant differences versus a chance level of 50% are indicated by asterisks (*p*-value for multiple comparisons, using Bonferroni correction for multiple comparisons, α level of 0.05 / 4 = 0.0125). (B) Individual data showing the percentage of preference for upright images. The horizontal axis represents age in days. The red circles and the blue squares represent the results for the intact and scrambled scene conditions, respectively. The red line is the regression line fitted to the red circles. The blue line is the regression line fitted to the blue squares.

between scenes and orientation ($F(1, 70) = 4.34$, $p = 0.041$, $\eta^2 = 0.01$), revealing that the total looking time for upright images was significantly longer than the inverted images in both intact and scrambled scene conditions. The main effects of age ($p = 0.41$, $\eta^2 = 0.00$) and scene ($p = 0.46$, $\eta^2 = 0.00$), two-way interaction between age and scene ($p = 0.81$, $\eta^2 = 0.00$), and three-way interaction ($p = 0.052$, $\eta^2 = 0.01$) were not significant.

We calculated individual percentage preference scores by dividing the infants' looking times for the image which contained the upright face (the upright image) during the test trial by the total looking time over the test trial (summation of looking time to the upright and inverted images) and then multiplying the ratio by 100. Fig 2A shows the mean preference for the upright image in the intact and scrambled conditions. To examine the interaction between participants' age and scene on infant's preference scores for the upright images, the preference scores were analyzed using a two-way ANOVA with participants' age (4–5, 6–7) as the between-participant factor and image condition (intact, scrambled) as the within-participants factor. The ANOVA yielded a significant main effect of participants' age, $F(1, 70) = 4.35$, $p = 0.041$, $\eta^2 = 0.04$, and image condition, $F(1, 70) = 4.61$, $p = 0.035$, $\eta^2 = 0.02$. The interaction was not significant, $F(1, 70) = 3.51$, $p = 0.065$, $\eta^2 = 0.02$.

To determine whether the infants in each age group discriminate between upright and inverted images and significantly look longer to upright images than inverted images, we conducted two-tailed one-sample *t*-tests versus chance level (50%) with a Bonferroni correction (α level of 0.05/4 = 0.0125) for multiple comparisons on the preference for the upright image in both the intact and scrambled scene conditions. For the 4- to 5-month-old infants, this analysis revealed that the preference score for the upright image was significantly greater than the chance level both in the intact scene condition ($M = 58.02\%$, SD = 8.05, $t(34) = 5.45$, $p < 0.001$, Cohen's $d = 1.00$) and the scrambled scene condition ($M = 57.70\%$, SD = 9.21, $t(34) = 4.58$, $p < 0.001$, Cohen's $d = 0.84$). Conversely, 6- to 7-month-old infants significantly preferred the upright image in the intact scene condition ($M = 56.98\%$, SD = 5.59, $t(36) = 6.38$, $p < 0.001$,

Cohen's $d$ = 1.25), but not in the scrambled scene condition ($M$ = 52.44%, SD = 8.34, $t$(36) = 1.60, $p$ = 0.12, Cohen's $d$ = 0.29).

Fig 2B represents individual data showing preference scores for the upright images plotted as a function of infants' age in days. To show developmental changes in infants' preference for the upright images in both the intact and scrambled scenes between 4 and 7 months, we calculated correlations between the preference for the upright images and age in days and tested the correlation coefficient. We found a significant negative correlation between age and preference score in the scrambled scene condition ($r$ = -0.25, $p$ = 0.031) but not in the intact scene condition ($r$ = -0.09, $p$ = 0.43).

In Experiment 1, we found that preference for the upright images was significantly higher than the chance level of 50% in the intact scenes in both 4- and 5-month-olds and 6- and 7-month-olds, revealing that infants over 4 months discriminate between the upright and inverted images and looked longer to the upright images containing the upright faces. These results are consistent with the previous findings that infants preferred upright faces [8–14].

More importantly, as we expected, we found developmental changes in the upright face preference in the scrambled scene condition, but not in the intact scene condition. That is, 4- to 5-month-old infants showed a preference for upright faces, whereas 6- to 7-month-old infants did not. These results suggest that scenic context would affect the preferences for the upright face in natural scenes in infants aged over 6 months. However, there remains a possibility that the lack of preference for the scrambled scene condition in 6- to 7-month-olds would be simply because they were more distracted by the low-level properties and/or complexity provided by the image scrambling (e.g., edges between blocks, etc.) than were 4- to 5-month-olds, and their attention would be pulled away from the upright faces of the display. To exclude this possibility, in Experiment 2 we directly investigated infants' spontaneous preferences for the scrambledness of images by simultaneously presenting an upright intact scene image and an upright scrambled scene image, in which faces were absent. Removing face frames from the stimuli allows us to examine infants' spontaneous preference for the image scrambledness itself. If we find no developmental change in preference for either the intact or scrambled image between 4- to 5-month-olds and 6- to 7-month-olds when faces did not appear, we could rule out the possibility that the scrambledness of the images were more distracting for infants' looking behavior in 6- to 7-month-olds than in 4- to 5-month-olds.

## Experiment 2

### Materials and methods

The methods for Experiment 2 were the same as those for Experiment 1 except for the following.

**Participants.**   The final sample in this experiment comprised thirty 4- to 5-month-old infants (14 females, $M$ age = 142.37 days; range = 114–164 days) and thirty 6- to 7-month-old infants (16 females, $M$ age = 200.67 days; range = 166–222 days). As in Experiment 1, the infants aged from 105 days to 164 days were assigned to the 4- to 5-month-old group, and infants aged from 165 days to 224 days were assigned to the 6- to 7-month-old group. They all were healthy and full-term Japanese infants. The study was conducted between February and August 2018.

We tested an additional 25 infants but excluded them from the analyses due to crying and/or fussiness (N = 7; 4–5 months: N = 4, 6–7 months: N = 3), a side bias greater than 90% (i.e., looking more than 90% of the time to one side) (N = 8; 4–5 months: N = 4, 6–7 months: N = 4), or premature birth (N = 1). This study was approved by a local ethical committee of Chuo University (2017–22, 2018–7).

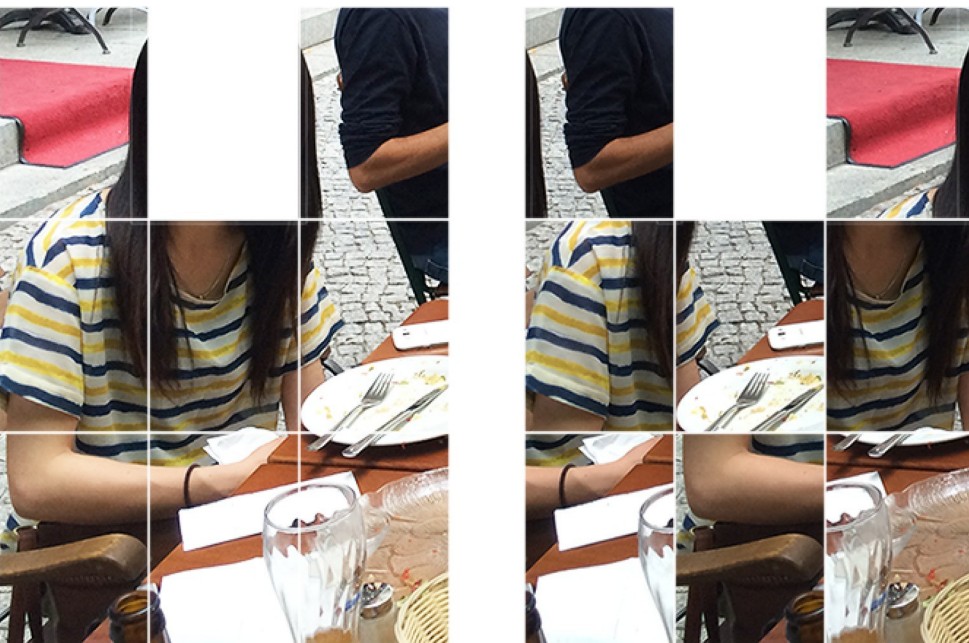

**Fig 3. Examples of stimuli used in Experiment 2.** Left: intact scene image from which the cell containing a face has been removed. Right: scrambled scene image from which the cell containing a face has been removed.

**Stimuli.** We used seven upright intact natural scene images and seven upright scrambled natural scene images, which were the same as those in Experiment 1. The cell containing a face was removed from each image for both intact and scrambled images, and this area was filled with homogeneous white (Fig 3).

**Procedure.** As in Experiment 1, infants were tested by a preferential-looking procedure composed of six 10-s trials. Three out of 7 natural scene images were randomly selected for each participant, and the intact and scrambled versions of the same three natural scenes were presented side by side on a monitor. We defined intact images without faces as targets and calculated preferential scores for the intact images for each infant. The position of the target was counterbalanced across participants across six trials.

Video recordings of infants' eye movements were analyzed frame-by-frame by an observer who recorded the total looking time for each intact and scrambled image in each trial. Also, a second independent observer analyzed 25% of the total data. Both observers were unaware of the position of the target (left or right) on the screen. Interobserver agreement (Pearson correlation), as computed on total fixation times for the infant and scrambled images across six trials, was $r = 0.93$, $p < 0.001$.

## Results and discussion

Table 2 indicates the mean total fixation times for the intact and scrambled images without a face in 4- and 5-month-olds and 6- and 7-month-olds. It seems that the mean total fixation times were not different between the intact and scrambled images in both age groups and there was no difference between age groups. To examine the effect of age (4–5, 6–7) and image type (intact, scrambled), we performed a two-way ANOVA on the mean total fixation time and found no significant main effects and interaction (age: $F(1, 58) = 1.30$, $p = 0.26$, $\eta^2 = 0.01$, image type: $F(1, 58) = 0.04$, $p = 0.84$, $\eta^2 = 0.00$, interaction: $F(1, 58) = 0.05$, $p = 0.82$, $\eta^2 = 0.00$).

**Table 2. Mean and standard deviation of total fixation time in 4- and 5-month-olds and 6- and 7-month-olds for the intact and scrambled images without a face.**

|  |  | Intact | | Scrambled | |
| --- | --- | --- | --- | --- | --- |
|  | N | M (s) | SD (s) | M (s) | SD (s) |
| 4–5 m | 30 | 25.24 | 4.28 | 25.22 | 3.54 |
| 6–7 m | 30 | 24.12 | 5.36 | 24.46 | 4.38 |

We calculated individual percentage preference scores by dividing the infants' looking time for the intact image without a face by the total looking time over the test trial (summation of looking time to the intact images and that to the scrambled images) and then multiplying the ratio by 100. Fig 4 represents the mean preference for the intact images without a face during the simultaneous presentation of intact and scrambled images. We analyzed preference scores for the intact image using an independent $t$-test to compare the preference score between 4- to 5-month-olds and 6- to 7-month-olds. This analysis revealed no significant change between age groups, $t(58) = 0.31$, $p = .75$, $d = 1.22$.

To determine whether the infants showed a significant preference for the intact images without a face, we conducted a two-tailed one-sample $t$-test (versus a chance level of 50%) on preference for the intact image. As a result, infants in all age groups did not show a significant preference for intact images (4- to 5-month-olds: $M = 49.92\%$, SD = 6.10, $t(29) = -0.07$, $p = 0.94$, Cohen's $d = -0.21$, 6- to 7-month-olds: $M = 49.39\%$, SD = 6.97, $t(29) = -0.48$, $p = 0.63$, Cohen's $d = -1.25$).

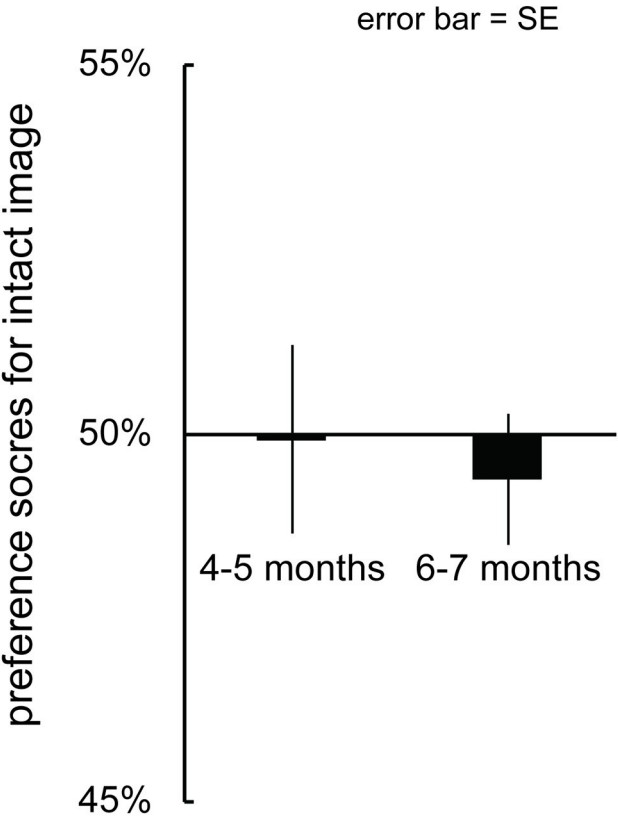

**Fig 4. Mean preference scores for the intact scene images without faces.** The error bars indicate mean standard error (SE).

Results of Experiment 2 revealed that infants of both age groups showed no preference for the scrambled images over the intact images when faces did not appear in the images. There was no developmental change in infants' preference, or lack thereof, for the scrambled images, implying that the low-level properties and/or stimulus complexity provided by image scrambledness did not affect infants' preference for the image in any age group.

There is the possibility that infants in both age groups showed no preference due to distraction by missing part of the stimuli. Because for Experiment 2 we used the images that a cell containing a face was simply removed and filled with homogenous white. We believe, however, that the missing part did not attract the infants' attention for the following reasons. First, previous studies revealed that infants preferred texture or pattern (dot, geometrics, etc.) rather than the unpatterned uniform region (e.g., [39, 40]). Second, the area removing the face cell was filled with white, which was the same color as the background, so that the area removing the face cell would integrate with the background.

## General discussion

In the present study, we explored the effect of scenic information on upright face preference in infancy. To this end, in Experiment 1, we tested visual preference for an upright over an inverted natural image containing a face in 4- to 7-month-old infants in two conditions: the intact scene condition, in which a face appeared in a natural intact scene, and the scrambled scene condition, in which a face appeared in a jumbled scene. We found that 4- to 5-month-old infants significantly preferred the upright images compared to the inverted images both in the intact scene and the scrambled scene. In contrast, 6- to 7-month-old infants showed a preference for the upright images only in the intact scene. In Experiment 2, we investigated infants' spontaneous preferences for scrambled images to exclude the possibility that 6- to 7-month-olds were more distracted by image scrambledness than 4- to 5-month-olds. To test this possibility, we removed the cell containing the face from upright intact and scrambled natural scene images and investigated visual preference for these two images presented simultaneously. Infants in both age groups showed no preference for either the scrambled images or the intact images when the faces did not appear. Additionally, there was no developmental change in infants' preferences for the scrambled images between 4- to 5-month-olds and 6- to 7-month-olds. To summarize, our results suggest that scenic contexts affected the upright face preference in infants over 6 months of age, whereas there was no such effect in infants under 5 months of age. These results revealed a developmental change in the effect of scenic context on face processing in a natural scene during the first year of life.

In this study, 6- to 7-month-olds preferred the upright face in the intact scene, although the upright face preference disappeared in the scrambled scene. This result suggests that the scenic context affects the preference for upright faces in infants aged 6 to 7 months. Results obtained from the 6- and 7-month-olds may be related to previous findings with adults reporting that scrambled scenes disrupt face processing [27]. Lewis and Edmonds [27] reported that face detection declined in the scrambled scene rather than in the intact scene, suggesting that adults use information from a natural scene in an early-attentive manner to find regions that are likely to contain a face. According to the multistage-processing model of face detection proposed by Lewis and Ellis [29], this early-attentive processing is the first stage of face detection. In this stage, the visual scene is scanned in a parallel and pre-attentive manner to help identify a candidate of location that may contain a face. This first pre-attentive processing is followed by template fitting and template evaluation, which involves the deformation of a face template through skew and rotation, and the evaluation of the degree of fit between the luminance pattern in the face candidate region and the deformed template. Considering the model of

naturalistic face detection [29], our results suggest that 6- to 7-month-old infants, as well as adults, may utilize scenic information as a cue to process faces within a whole scene. For infants aged over 6 months who develop an adult-like face-detection system, the scrambled scene might disrupt the identification of the candidate location of a face and lead to difficulty finding the face. Therefore, 6- and 7-month-old infants might not show a significant preference for upright images in the scrambled scene.

In contrast to 6- to 7-month-olds, 4- to 5-month-old infants preferred an upright face both in the intact and scrambled scenes, indicating that young infants have sensitivity to the presence of the face despite the presence or absence of a scenic context. This is in accord with previous findings consistently reporting infants' preferences for upright faces over inverted ones. By at least 3 months of age, infants exhibit a preference for upright photographic faces based on a face-specific processing mechanism [8, 12]. Also, recent evidence has revealed that infants prefer to look at a face in more complex visual displays. By using the visual search paradigm, previous studies have indicated that infants over 4 months of age looked longer at a face among multiple distractors of non-face objects [17–19]. In addition, when investigating eye movements to faces in complex scenes in 3-, 6-, and 9-month-olds during free viewing of animated films, infants' attentive looking at the face gradually increased between 3 and 9 months of age [20, 21]. These previous studies demonstrated that at 4 months of age, infants can process faces embedded in a complex scene. The current study extends these previous findings and shows that younger infants ignore the scenic information during face processing. Given that a template or representation of the human face has already developed a few months after birth [1], infants under 5 months of age may show a preference for upright faces based only on the face template.

Our results imply that the effect of scenic information on face processing seems to be more salient with age. Results of the *t*-tests versus chance level revealed that 4- and 5-month-old infants significantly prefer the upright face in both the intact and scrambled scenes, and 6- to 7-month-olds significantly prefer the upright face only in the intact scene. This developmental change in the visual preference in the scrambled scene condition was corroborated by a significant negative correlation showing the decrease in the upright face preference between 4 and 7-month-olds. The developmental trajectory for face preference in a natural scene observed in the current study suggests that infants aged at around 5 and 6 months are in a transitional stage from "just processing a face" regardless of the scene to processing a face based upon an analysis of the entire visual image. In one study, the developmental change of face processing ability, known as the perceptual narrowing phenomenon, also occurred at similar ages. Kelly and colleagues [41] showed that infants' ability to differentiate individuals of other-race faces gradually declines between 3 and 9 months of age, suggesting that an innate sensitivity to faces is altered gradually through perceptual learning from environmental input. Considering the evidence of perceptual narrowing, the template or representation of faces would become more sophisticated with increased age and accumulation of visual experience. The developmental change in the upright face preference within a natural scene may be related to the alteration in face processing by visual input from the environment.

The development of face preference in a natural scene may be associated with the early development of body processing. Lewis and Ellis [29] pointed out that it is most likely to see a face above a body and a pair of shoulders, and this context in which a face appears may affect face processing in adults. This suggests that infants may need knowledge about the human body to show a preference for upright faces in a natural scene. Prior studies demonstrated that infants acquired some extent of knowledge about the human body by 5 months of age. Zieber and colleagues [33] found that at 3.5 months of age, infants were sensitive to the characteristics of the human body. Infants aged 3.5 months also exhibited discrimination both between intact

bodies and those with parts in wrong locations and between intact bodies and those with distorted part proportions. Additionally, the holistic processing of the body develops by 5 months of age [34]. When investigating 5- and 9-month-old infants' discrimination of body posture in the context of the whole body, isolated body parts, or scrambled body, infants over 5 months showed successful discrimination of body posture only in the context of the whole body. Considering these findings, acquiring knowledge of human body structure and holistic body processing by 5 months of age might play an important role in face processing in natural scenes in infants.

We found the effect of scenic context on infants' face processing in 6- and 7-month-olds, but not in 4- and 5-month-olds. However, we limit our conclusions to face preference and cannot mention how the scenic context affects each stage of face processing, from an early attentive stage of face detection to the visual preference measured by looking behavior. An eye-tracking study may provide a vision of the future directions in the effect of scrambled scenes on face processing in infancy. Also, we cannot fully deny the possibility that infants' preference for the upright images in the intact scene was induced not only by the upright faces but also by the upper half of the human bodies, although we believe the upright human body preference may have no or little effect on the results of the current study for the following reasons. First, a previous study reported infants over 3 months showed long fixation only to faces and very few to bodies in natural scenes [23]. Second, if the upper half of the body induced infants' preferences, infants should show a preference for the intact images without faces than for the scrambled images without faces in Experiment 2. As a result, we did not any preference. However, an important next step may be to investigate the effect of the upper half of bodies on infants' face preference.

## Conclusions

The current study explored the effect of natural scenic contexts on infants' preferences for upright faces in 4- to 7-month-old infants by comparing their preferences between intact and scrambled scenes. We revealed that the ability to utilize scenic information to process faces develops at 6 to 7 months of age. Also, there may be a developmental trajectory with respect to utilizing scenic information between 5 and 6 months of age: this ability may become more robust at 6 months of age. Although previous studies with infants have already demonstrated the upright face preference without scenic contexts, our study is the first report showing the effects of a naturalistic scene context on infants' preferences for upright faces.

## Supporting information

**S1 Data. Raw data of infants' looking time for Experiment 1.**
(XLSX)

**S2 Data. Raw data of infants' looking time for Experiment 2.**< /SI_Caption>
(XLSX)

## Acknowledgments

The authors thank Hiroko Ichikawa, Kazuki Sato, Kohsuke Murakami, Shuma Trusumi, Yuta Ujiie, and Jiale Yang for their help in data collection. We specially thank the infants and their parents for their kindness and cooperation in the research.

## Author Contributions

**Conceptualization:** Megumi Kobayashi, So Kanazawa, Masami K. Yamaguchi.

**Data curation:** Megumi Kobayashi.

**Formal analysis:** Megumi Kobayashi.

**Funding acquisition:** Megumi Kobayashi, Masami K. Yamaguchi.

**Investigation:** Megumi Kobayashi.

**Methodology:** Megumi Kobayashi, So Kanazawa.

**Project administration:** Masami K. Yamaguchi.

**Supervision:** So Kanazawa, Masami K. Yamaguchi.

**Visualization:** Megumi Kobayashi.

**Writing – original draft:** Megumi Kobayashi.

**Writing – review & editing:** Megumi Kobayashi, So Kanazawa, Masami K. Yamaguchi.

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
