## [Decision Letter · Decision Letter 0]

28 Feb 2023

PONE-D-22-32054The role of scenic context on upright face preference in infancyPLOS ONE

Dear Dr. Kobayashi,

Thank you for submitting your manuscript to PLOS ONE. After careful consideration, we feel that it has merit but does not fully meet PLOS ONE’s publication criteria as it currently stands. Therefore, we invite you to submit a revised version of the manuscript that addresses the points raised during the review process. Both reviewers provide specific comments on the methods and analyses used in the study, with suggestions for clarification and improvement. Reviewer 1 notes the need for clearer research questions and organisation of the introduction, consideration of age as a continuous variable, and inclusion of limitations and future directions in the discussion. Reviewer 2 points out potential confounding factors in the interpretation of the results, the need for clearer explanations of the relationship between contextual information and upright preference, and discrepancies between text and figure illustrations.

Having made my own assessment of the manuscript, I agree that it would benefit greatly from a clearer statement of purpose and a more tailored introduction. Finally, please also check the manuscript carefully for typing errors.

We look forward to receiving your revised manuscript.

Kind regards,

Anna Sagana, PhD

Academic Editor

PLOS ONE

Journal Requirements:

Reviewers' comments:

Reviewer's Responses to Questions

**Comments to the Author**

1. Is the manuscript technically sound, and do the data support the conclusions?

Reviewer #1: Partly

Reviewer #2: Partly

2. Has the statistical analysis been performed appropriately and rigorously? 

Reviewer #1: Yes

Reviewer #2: Yes

3. Have the authors made all data underlying the findings in their manuscript fully available?

Reviewer #1: Yes

Reviewer #2: Yes

4. Is the manuscript presented in an intelligible fashion and written in standard English?

Reviewer #1: No

Reviewer #2: Yes

5. Review Comments to the Author

Reviewer #1: The manuscript under review investigates the role of scenic information in face processing in infants between 4 and 7 months of age. In two cross-sectional experiments, the authors show developmental differences in how scenic information influences how infants process faces. In particular, contextual features seem to influence visual preference in older (i.e., 6-7 months) but not younger (i.e., 4-5 months) infants. The authors discuss this difference in relation to previous findings on perceptual narrowing.

Overall, the study design is sound and on a topic that is of interest for the field. I have a few comments on the organization of the manuscript, as well as on the analyses.

A good part of the introduction discusses several studies without providing a clear indication of the research questions of the current work, which are described only at the end. Moreover, the first three paragraphs describe the visual preference phenomenon that are marginally related to the topic of the current study. I would recommend to clearly state the research question(s) and extensively discuss the relevant literature (the role of contextual factors in face processing), with only minimal reference to the general development of face processing. Several of these studies are reported in the discussion.

In both experiments, the authors split the sample into two groups. What were the exact criteria used for this? Do the authors consider the small age differences between groups as meaningful? Would considering age as a continuous variable be more meaningful for the goal of the study?

Why wasn’t age considered as a factor in the analyses on the total fixation times of experiment 1?

A p-value of .065 is nonsignificant and should be considered as such.

Since a fixed duration was chosen for the stimuli presentation, were there any differences in the total looking time across age groups?

Can the authors clarify the reasons to analyze the preference scores for upright images using both an anova and one-sample t-tests? There is also minimal discussion of the results vs. chance level.

Could the authors further elaborate whether the missing piece in the stimuli used in Experiment 2 could have attracted the infant's attention, thus influencing the results? Why not use a new set of images without faces to test age differences in the preference for textures (intact vs. scrambled)?

Overall, since the study has a cross-sectional design, I would suggest referring to any age-related results in terms of “difference” more than “change”.

The discussion would benefit from the inclusion of the limitations of the study and future directions.

Several typos are left throughout the manuscript.

Reviewer #2: The manusciprt “The role of scenic context on upright face preference in infancy” provided interesting insights into the role of scenic information in face processing and how it is used by infants. The finding that infants aged 6 months or more preferred upright faces in an intact scene, but not in a scrambled scene, indicates that they are sensitive to the surrounding information in face perception. The study also ruled out the potential confounding factor of more distraction by image scramble in 4- to 5-month-olds. Overall, this study highlights the importance of considering the role of surrounding information in face processing, and suggests that infants are able to utilize such information from a relatively young age.

I have several comments as follows:

1. Robust upright image preference over inverted ones was reported in 4- to 5-month-olds, no matter the surrounding information was intact (meaning physically aligned with the face) or scrambled. However, the upright preference may not only be driven by the differences in face directions, but also by the direction differences in other image parcels. For instance, the upright image of the human body may have more meaning than the inverted version, even in the scrambled condition. Thus, “the preference score for upright images” could not reflect pure face perception per se, but reflect person and scene perception mixed together. The authors should resolve the issue as it may lead to other interpretations about the missing upright preference in infants aged 6 months or more.

2. Why would the scrambled contextual information diminish the upright preference in 6-month-olds? The authors mentioned a parallel and pre-attentive manner of visual scene scanning to help face detection (Page 24). I did not follow the logic here. The author should address this more clearly.

3. Line 280: “We found a significant negative correlation between age and preference score in the intact scene condition (r = -0.25, p < 0.05) but not in the scrambled condition (r = -0.09, n.s.). ” Inconsistent descriptions with figure illustration. Regression seems to be more significant in the scrambled condition according to Figure 2b.

4. Line 199: Three images out of 7 images were randomly selected for each participant in the intact scene condition. I am curious about the purpose of this procedural design.

6. PLOS authors have the option to publish the peer review history of their article (what does this mean?). If published, this will include your full peer review and any attached files.

Reviewer #1: **Yes: **Stefania Conte

Reviewer #2: No

---

## [Author Response · Author response to Decision Letter 0]

13 Apr 2023

Reviewer #1: The manuscript under review investigates the role of scenic information in face processing in infants between 4 and 7 months of age. In two cross-sectional experiments, the authors show developmental differences in how scenic information influences how infants process faces. In particular, contextual features seem to influence visual preference in older (i.e., 6-7 months) but not younger (i.e., 4-5 months) infants. The authors discuss this difference in relation to previous findings on perceptual narrowing.

Overall, the study design is sound and on a topic that is of interest for the field. I have a few comments on the organization of the manuscript, as well as on the analyses.

A good part of the introduction discusses several studies without providing a clear indication of the research questions of the current work, which are described only at the end. Moreover, the first three paragraphs describe the visual preference phenomenon that are marginally related to the topic of the current study. I would recommend to clearly state the research question(s) and extensively discuss the relevant literature (the role of contextual factors in face processing), with only minimal reference to the general development of face processing. Several of these studies are reported in the discussion.

Response- Thank you for your suggestion. According to your suggestion, we revised the Introduction section to clarify our research question. (pp. 3-7)

In both experiments, the authors split the sample into two groups. What were the exact criteria used for this? Do the authors consider the small age differences between groups as meaningful? Would considering age as a continuous variable be more meaningful for the goal of the study?

Response- As we explain in the Participants section, the current study adopted the criteria to split the participants from the previous infant studies (e.g., Imura et al., 2006; Tsuruhara et al., 2010; Ichikawa et al., 2014; Ujiie et al., 2018). We calculated the range of age in days for an age group in months as follows:

 The minimum age in days = 30× j -15

 The maximum age in days = 30× j + 14

 (Note: “j” is the age in the month)

Therefore, the 4-month-old group contains infants aged between 105 days and 134 days (mean age is approximately 120) and the 5-month-old group contains infants aged between 135 days and 164 days (mean age is approximately 150). Infants aged between 165 days and 194 days (mean age is approximately 180) were assigned to the 6-month-old group, and infants aged between 195 days and 224 days (mean age is approximately 210) were assigned to the 7-month-old group.

We believe that considering age both as a categorical variable (e.g., 4-5 month-group, 6-7 month-group) and a continuous variable is important and meaningful. To examine whether infants in a specific age group discriminate between two images and significantly prefer one image of them, we need to consider age as the categorical variable. We split infants into two age groups, calculated each group's mean preference, and performed the statistical analysis versus chance level on mean preference score (Figure 2A). On the other hand, to examine a gradual change with increased age, it would be more suitable to consider age as the continuous variable. In the current study, we considered age in days as the continuous variable and tested the correlation coefficient between preference scores and infants’ age in days (Figure 2B).

Why wasn’t age considered as a factor in the analyses on the total fixation times of experiment 1?

Response- According to your suggestion, we conducted an ANOVA with three factors (age, scene, and orientation) for the total fixation times of Experiment 1 (pp. 12, line 207 – pp. 13, line 220) and an ANOVA with two factors (age and image type) for the total fixation times of Experiment 2 (pp.20, lines 335 – 338). We revised the Results section of both experiments. 

A p-value of .065 is nonsignificant and should be considered as such.

Response- We revised the texts in the Results section mentioning the interpretation of marginal significance (pp. 14, line 235). 

Since a fixed duration was chosen for the stimuli presentation, were there any differences in the total looking time across age groups?

Response- A preferential-looking method requires us to fix the stimulus duration that two stimuli were presented and calculate the percentage of looking time for one stimulus as preference scores. We conducted a 3-way ANOVA with age (4-5mo, 6-7mo), orientation (upright, inverted), and scene (intact, scrambled) on infants’ looking time observed in Experiment 1 and did not find the main effect of age, revealing that there was no difference in total looking time for upright and inverted stimuli between age groups. But we found a significant two-way interaction between age and orientation, with longer looking time for the upright images in 4-5-month-olds than in 6-7-month-olds. This was reflected by 4-5-month-olds’ significant preference for the upright images both in the intact and scrambled scene conditions and 6-7-month-olds’ significant preference only in the intact condition.

Can the authors clarify the reasons to analyze the preference scores for upright images using both an anova and one-sample t-tests? There is also minimal discussion of the results vs. chance level.

Response- The reason we conducted an ANOVA with age and condition was that we predicted an interaction between age and condition. We conducted t-tests (vs. chance level of 50%) to confirm whether infants’ fixation time was significantly biased to the upright images. No statistical significance versus chance level (50%) means that infants did not discriminate between the upright and inverted images and did not show any preference for the upright image. We added the clarified reasons and more discussion about the results of t-tests (pp. 12, line 206 – pp. 16, line 271). 

Could the authors further elaborate whether the missing piece in the stimuli used in Experiment 2 could have attracted the infant's attention, thus influencing the results? Why not use a new set of images without faces to test age differences in the preference for textures (intact vs. scrambled)?

Response- In Experiment 2 we believe that infants were not attracted by the area of missing pieces in the stimuli (white area), then focused on the scene for the following reasons. First, previous studies have consistently reported that infants preferred texture or pattern (dot, geometrics, etc.) rather than unpatterned uniform region (e.g., Fantz, 1963; Fantz &Yeh, 1979). Second, the area removing the face cell was filled with homogeneous white that was the same color as the background, so that the area removing the face cell would integrate with the background. We added these points to the Results and Discussion section of Experiment 2 (pp. 22, lines 368 – 375).

 In Lewis & Edmonds (2003), to remove a face cell from the scene, a cell from the bottom of the image was duplicated and pasted to cover the face cell. If applying this image transformation, low-level properties and/or complexity would emerge between the pasted cell and the other cells surrounding it in the intact scene image as well. Therefore, we used the images that a face cell was simply removed.

Overall, since the study has a cross-sectional design, I would suggest referring to any age-related results in terms of “difference” more than “change”.

Response- According to your suggestion, we revised the manuscript.

The discussion would benefit from the inclusion of the limitations of the study and future directions.

Response- According to your suggestion, we added the limitation and future directions to the Discussion section (pp. 28, line 269 – pp. 29, line 483).

Several typos are left throughout the manuscript.

Response- We had our manuscript proofread.

Reviewer #2: The manusciprt “The role of scenic context on upright face preference in infancy” provided interesting insights into the role of scenic information in face processing and how it is used by infants. The finding that infants aged 6 months or more preferred upright faces in an intact scene, but not in a scrambled scene, indicates that they are sensitive to the surrounding information in face perception. The study also ruled out the potential confounding factor of more distraction by image scramble in 4- to 5-month-olds. Overall, this study highlights the importance of considering the role of surrounding information in face processing, and suggests that infants are able to utilize such information from a relatively young age.

I have several comments as follows:

1. Robust upright image preference over inverted ones was reported in 4- to 5-month-olds, no matter the surrounding information was intact (meaning physically aligned with the face) or scrambled. However, the upright preference may not only be driven by the differences in face directions, but also by the direction differences in other image parcels. For instance, the upright image of the human body may have more meaning than the inverted version, even in the scrambled condition. Thus, “the preference score for upright images” could not reflect pure face perception per se, but reflect person and scene perception mixed together. The authors should resolve the issue as it may lead to other interpretations about the missing upright preference in infants aged 6 months or more.

Response- Although we could not provide a direct answer to the question of whether upright image preference in 4-5-month-olds was driven not only by the upright face but also by the upright human body because of no eye-tracking data, we believe that the upright human preference had no or little effect on the upright image preference in the current study as following reasons. First, if the upper half of the body induced the upright image preference in 4-5-month-olds, they should look longer to the intact images without faces (the upper half of the body) than to scrambled images without faces in Experiment 2. However, we did not find any preference. Second, a previous study reported that even if the whole body was presented, just faces, not bodies, provoked infants’ fixation, or preference (Figure S3a in Kelly et al., 2019). 

However, we cannot fully deny the possibility that infants’ preferences might reflect the responses of both faces and bodies. We added this point to the Discussion section as limitations and future directions. (pp. 28, line 474 – pp. 29, line 483)

2. Why would the scrambled contextual information diminish the upright preference in 6-month-olds? The authors mentioned a parallel and pre-attentive manner of visual scene scanning to help face detection (Page 24). I did not follow the logic here. The author should address this more clearly.

Response- According to your suggestion, we added some explanations in the Discussion section to clarify the logic. (pp. 24, line 413 – pp. 25, line 416) 

3. Line 280: “We found a significant negative correlation between age and preference score in the intact scene condition (r = -0.25, p < 0.05) but not in the scrambled condition (r = -0.09, n.s.). ” Inconsistent descriptions with figure illustration. Regression seems to be more significant in the scrambled condition according to Figure 2b.

Response- Thank you for your point out. We reported mistaken statistical values for the correlation coefficient and revised them. (pp. 16, lines 264 – 266) 

4. Line 199: Three images out of 7 images were randomly selected for each participant in the intact scene condition. I am curious about the purpose of this procedural design.

Response- When the developmental studies with infants examined infants’ preference for faces or discrimination between faces, a few stimuli sets are commonly used. These studies present faces on a homogenous simple background. In contrast, for the present study that examined the effect of scene on face processing, we considered we need various scene images containing a face. Therefore, we used 7 images of various scenes. Although it might be best that all 7 images were presented for each infant, it was difficult to do that due to infant-specific constraints (e.g., fussiness, difficulty sustaining attention to the monitor, etc.). So, we randomly choose 3 images out of 7 for each infant and investigated averaged infants’ response to 7 images across participants.

---

## [Editor Report · Decision Letter 1]

10 May 2023

PONE-D-22-32054R1The role of scenic context on upright face preference in infancyPLOS ONE

Dear Dr. Kobayashi,

Thank you for submitting your manuscript to PLOS ONE. After careful consideration, we are please to see that all reviewers' comments have been successfully addressed. Your efforts in improving the paper are highly appreciated.  However, before we can accept your paper for publication, we need you to report exact p-values for all values greater than or equal to 0.001 as per journal guidelines. Accurate reporting of p-values is essential for ensuring the reproducibility and validity of research findings. Therefore, I kindly ask you to submit a revised version with the exact p-values. 

We look forward to receiving your revised manuscript.

Kind regards,

Anna Sagana, PhD

Academic Editor

PLOS ONE
---

## [Author Response · Author response to Decision Letter 1]

12 May 2023

Thank you for submitting your manuscript to PLOS ONE. After careful consideration, we are please to see that all reviewers' comments have been successfully addressed. Your efforts in improving the paper are highly appreciated. 

However, before we can accept your paper for publication, we need you to report exact p-values for all values greater than or equal to 0.001 as per journal guidelines. Accurate reporting of p-values is essential for ensuring the reproducibility and validity of research findings. Therefore, I kindly ask you to submit a revised version with the exact p-values.

Response- According to your suggestion, we have revised the p-values to report exact values for all greater than or equal to 0.001.

---

## [Editor Report · Decision Letter 2]

22 Jun 2023

The role of scenic context on upright face preference in infancy

PONE-D-22-32054R2

Dear Dr. Kobayashi,

We’re pleased to inform you that your manuscript has been judged scientifically suitable for publication and will be formally accepted for publication once it meets all outstanding technical requirements.

Kind regards,

Anna Sagana, PhD

Academic Editor

PLOS ONE
---

## [Editor Report · Acceptance letter]

5 Jul 2023

PONE-D-22-32054R2 

The role of scenic context on upright face preference in infancy 

Dear Dr. Kobayashi:

I'm pleased to inform you that your manuscript has been deemed suitable for publication in PLOS ONE. Congratulations! Your manuscript is now with our production department. 

Kind regards, 

on behalf of

Dr. Anna Sagana 

Academic Editor

PLOS ONE